# Isothermal Oxidation Performance of Laser Cladding Assisted with Preheat (LCAP) Tribaloy T-800 Composite Coatings Deposited on EN8

Sipiwe Trinity Nyadongo [1], Sisa Lesley Pityana [2,]* and Eyitayo Olatunde Olakanmi [1]

1 Mechanical, Energy, Industrial and Manufacturing Engineering, Botswana International University of Science and Technology, Palapye 00267, Botswana; sipiwe.nyadongo@studentmail.biust.ac.bw (S.T.N.); olakanmie@biust.ac.bw (E.O.O.)
2 Council for Scientific and Industrial Research, Pretoria 0001, South Africa
* Correspondence: spityana@csir.co.za; Tel.: +27-833-168-690

**Abstract:** It is anticipated that laser cladding assisted with preheat (LCAP)-deposited Tribaloy (T-800) composite coatings enhances resistance to structural degradation upon exposure to elevated-temperature oxidation service environments. The oxidation kinetics of LCAP T-800 composite coatings deposited on EN8 substrate and its mechanisms have not been explored in severe conditions that are similar to operational parameters. The isothermal oxidation behaviour of the T-800 composite coating deposited on EN8 via LCAP was studied at 800 °C in air for up to 120 h (5 × 24 h cycles) and contrasted to that of uncoated samples. The mass gain per unit area of the coating was eight times less than that of the uncoated EN8 substrate. The parabolic rate constant $(K_p)$ for EN8 was $6.72 \times 10^{-12}$ $g^2 \cdot cm^{-4} \cdot s^{-1}$, whilst that for the T-800 composite coating was $8.1 \times 10^{-13}$ $g^2 \cdot cm^{-4} \cdot s^{-1}$. This was attributed to a stable chromium oxide ($Cr_2O_3$) layer that formed on the coating surface, thereby preventing further oxidation, whilst the iron oxide film that formed on the EN8 substrate allowed the permeation of the oxygen ions into the oxide. The iron oxide ($Fe_2O_3$) film that developed on EN8 spalled, as evidenced by the cracking of oxide when the oxidation time was greater than 72 h, whilst the $Cr_2O_3$ film maintained its integrity up to 120 h. A parabolic law was observed by the T-800 composite coating, whilst a paralinear law was reported for EN8 at 800 °C up to 120 h. This coating can be used in turbine parts where temperatures are <800 °C.

**Keywords:** high-temperature oxidation; laser cladding assisted with preheat (LCAP); Tribaloy T-800 composite; EN8 steel; coating





## 1. Introduction

Oxidation occurs when metals/alloys are exposed to oxygen/air [1]. It also results in the formation of porous and noncoherent oxides, which leads to the failure of the component [2]. The dispersion of oxygen on the equipment surfaces leads to oxidation, which then propagates into the core of the material. Equipment operating at high temperatures is prone to this phenomenon, which often leads to premature equipment degradation and consequent failure. A thin oxide layer well bonded to the surface of the coating and slow in growth will offer good protection to the substrate from additional oxidation. Various coatings can be used to subvert the oxidation phenomenon. For instance, Vasudev et al. [2] explored the use of superalloys (Inconel 718) for enhancing the elevated-temperature oxidation resistance of turbocharger housing. Inconel 718 exhibited resistance to high-temperature oxidation and erosion due to the formation of protective oxides such as nickel chromite ($NiCr_2O_4$), chromium oxide ($Cr_2O_3$) and aluminium oxide ($Al_2O_3$) at an elevated temperature of 900 °C. Feng et al. [3] also investigated the high-temperature oxidation resistance of laser-cladded coatings and concluded that protective oxides prevented the diffusion of oxygen into the coating layer. Peng et al. [4] studied the oxidation behaviour of

Tribaloy T-400 and Tribaloy T-800 at 900 °C for up to 1000 h. It was discovered that Tribaloy T-800 offers enhanced oxidation resistance compared to Tribaloy T-400. This was attributed to its high Cr content, which readily forms a $Cr_2O_3$ layer, thereby protecting the coating. Furthermore, the depth of penetration of the oxides for Tribaloy T-800 was less than that for Tribaloy T-400, and the former attained steady-state oxidation quicker than the latter.

Laser cladding (LC) is a remanufacturing technique that aids in restoring worn-out/end-of-life (EOL) parts to "as new". It uses additive manufacturing principles, where a laser beam irradiates the substrate before the addition of metal powder, wire or paste, which then forms a coating. Laser cladding has several advantages, including the formation of a strong metallurgical bond, low heat-affected zone (HAZ), low dilution ratio and good surface finish, amongst others [5,6]. However, it has some undesired characteristics, such as crack formation and porosity, that can form due to high-temperature gradients or mismatched thermophysical properties [7]. These defects affect the performance of the coatings during service, as they could be initiation points to propagate corrosion (high-temperature oxidation) and/or wear. Hence, these defects are mitigated by adopting substrate preheat, amongst other methods [8–11]. A technique of substrate preheating, in which the substrate is heated via a hot plate to a specified temperature before the deposited material and the substrate are irradiated with a laser beam, is designated laser cladding assisted with preheat (LCAP). This research adopted LCAP for the fabrication of a T-800/WC composite coating for application in high-temperature environments.

Each manufacturing method is unique in how it influences the processing–microstructure–property relationship of products because the process parameter–material interactions vary for different manufacturing techniques. This accounts for an alteration in the performance of products fabricated with different manufacturing techniques when evaluated in the same service environments. This concept has been demonstrated in the work of Pala et al. [12], Tuominen et al. [13] and Khanna et al. [14]. Isothermal oxidation studies of T-800 or its composites deposited on substrate materials such as stainless steel 304 and 310L via LC, HVOF and plasma spray techniques have been explored [15]. EN8 is an unalloyed carbon steel with reasonable tensile strength. It is used in fabricating general engineering components, which find applications in elevated-temperature service environments. To the best of our knowledge, there has been no study that elucidates the nature and variants of environmental degradation encountered by an uncoated EN8 substrate and EN8 coated with a T-800 composite coating manufactured via the LCAP technique when subjected to an elevated-temperature service environment. Without such study, it is difficult to ascertain relevant preventive measures that should be implemented against materials loss and failures during the isothermal oxidation of an LCAP-deposited composite coating on EN8 substrate if the safety and reliability of the coating/substrate system and LCAP method will be guaranteed for industrial users. This research provides information on the comparative analysis of the isothermal oxidation behaviour of an uncoated EN8 substrate and EN8 coated with an LCAP-fabricated T-800 composite coating at a service temperature of 800 °C by exploring sample mass over time, phase analysis of oxide scales, observation of morphology and microstructure of oxide scales and oxidation kinetics. Outcomes from this study were employed in elucidating the possible mechanism of oxidation of the coated substrate at 800 °C.

## 2. Materials and Methods

Tribaloy T-800/WC composite (80 wt.% Tribaloy T-800 + 20 wt.% tungsten carbide (WC-86)) coatings were deposited on EN8 substrate using LCAP. The chemical compositions of T-800 (coating material), WC-86 (reinforcing material) and EN8 stainless steel (substrate) supplied by Kennametal (Pty.) Ltd. (Pittsburg, Pennyslavia) are given in Table 1.

**Table 1.** Chemical composition of tungsten carbide WC-86 and Tribaloy T-800.

| Chemical Composition | Co | Mo | Cr | Si | C | Ni | Fe | WC | Mn | P | S |
|---|---|---|---|---|---|---|---|---|---|---|---|
| Tribaloy T-800 | 46.7 (Bal.) | 28.5 | 18 | 3.5 | 0.8 | 1.5 | 1 | - | - | - | - |
| Tungsten Carbide (WC-86) | 10 | - | 4 | - | - | - | - | 86 | - | - | - |
| EN8 Stainless Steel | - | - | - | 0.25 | 0.4 | - | 98.52 (Bal.) | - | 0.8 | 0.015 | 0.015 |

LCAP was employed in this study to mitigate crack formation, which is associated with laser cladding (LC). LCAP deposition was implemented by adopting a laser power of 1750 W, a 4 mm spot diameter, a substrate preheat temperature of 400 °C and an LLED of 50 kJ/m in the deposition of the T-800/WC coating. LLED is a ratio of laser power and scan velocity. For instance, 50 kJ/m was obtained by dividing a laser power of 1750 W by a scan velocity of 2.1 m/min.

Fabricated specimens with a surface area of approximately 272 mm$^2$ were prepared. SiC abrasive paper was used for conventionally grinding the samples before cleaning in an acetone bath for 15 min to eliminate impurities [16]. The specimens were dried and weighed [17]. A Mettler Toledo digital balance with a sensitivity of 0.0001 g was used to weigh the specimens before and after the oxidation test. Isothermal oxidation tests were carried out in a no-name air furnace (still air) at 800 °C for 5 × 24 h cycles. The experiments were repeated twice for each set of parameters for both the uncoated (EN8) and coated (T-800/WC) specimens. Thereafter, the average weight gain was calculated. The samples were placed in crucibles individually before being loaded simultaneously into the furnace. However, they were unloaded from the furnace after 24, 48, 72, 94 and 120 h cycles. The specimens were then allowed to cool to room temperature in a desiccator to prevent moisture gathering on each specimen during cooling. Thereafter the specimens were weighed to determine the change in mass [18]. The mass gain per unit area $M_a$ was calculated according to Equation (1) [19].

$$M_a = \frac{M_f - M_i}{A} \qquad (1)$$

where $M_a$ is the mass gain per unit area, $M_f$ is the final mass after the oxidation test, $M_i$ is the initial mass before the oxidation test and $A$ is the total surface area (before the oxidation test). The parabolic rate constant ($K_p$) is a measure of the oxidation resistance of the coating. The lower the value of $K_p$, the higher the resistance to elevated-temperature oxidation. $K_p$ was calculated using Equation (2).

$$K_p = \frac{M_a{}^2}{t} \qquad (2)$$

where $K_p$ the parabolic rate constant, t is the time of specimen exposure in the furnace and $M_a$ is the mass gained per unit area. The oxidised specimens were further analysed using a D8 Advance Bruker with Davinci Design XRD machine (with monochromatic CuK$\alpha$ radiation, Bruker, Billerica, MA, USA) to identify the phases developed in the oxidised (coated) specimens. A JEOL JSM-7100F field emission scanning electron microscope (SEM, JEOL, Tokyo, Japan) with an energy dispersive spectroscopy (EDS) feature was used to observe the top-view (morphology) and in-cross-section (microstructure of oxide scales), as well as the chemical composition, of the formed oxide scale.

## 3. Results and Discussion

*3.1. Gravimetric Analysis of the Oxidation Behaviour of the EN8 Substrate and T-800/WC Coating at 800 °C*

Gravimetric analysis was used for the comparison of oxidation behaviour between the EN8 substrate and T-800/WC coating. The variation in mass gain per unit surface area

with oxidation time (24, 48, 72, 96, 120 h) at 800 °C for the uncoated EN8 substrate and the T-800/WC coating is shown in Figure 1. It is discernible from Figure 1 that the weight gain per unit area for both the uncoated substrate and the coated (T-800/WC) substrate increases as the oxidation time increases. The weight gain per unit area of the coated substrate is lower than that of the uncoated substrate material [20]. This can be observed from the weight gain per unit area of 3.16, 4.71 and 6.61 mg/mm$^2$ for EN8, whilst for the Tribaloy T-800/WC coating, it is 1.51, 2.46 and 3.13 mg/mm$^2$ at 24, 48 and 72 h, respectively. In Figure 1, the weight gain for the EN8 substrate is first seen to be parabolic with increasing temperature. However, at time 72 > t > 96 h (breakaway section), there is a rapid leap in the mass gain of EN8 from 6.61 to 26.85 mg/mm$^2$, almost four times more weight gain than the coated specimen. For the T-800/WC coating, the weight gain is quicker when the oxidation time is <72 h compared to >72 h. Quicker weight gain noticed when the oxidation time is <72 h could be attributed to the exposure of the surface of the samples to air without any form of protection. Thereafter (t > 72 h), the formed protective oxide layer reduces the oxidation of the coating. Figure 1 shows that the trend of oxide growth on the T-800/WC coatings follows a parabolic law, which is the normal oxidation trend for alloys and composites. These findings are similar to those of Buscail et al. [21], who investigated the isothermal behaviour of a cobalt-based alloy at 800–1000 °C, where a parabolic rate law was observed in the mass gain per unit surface area.

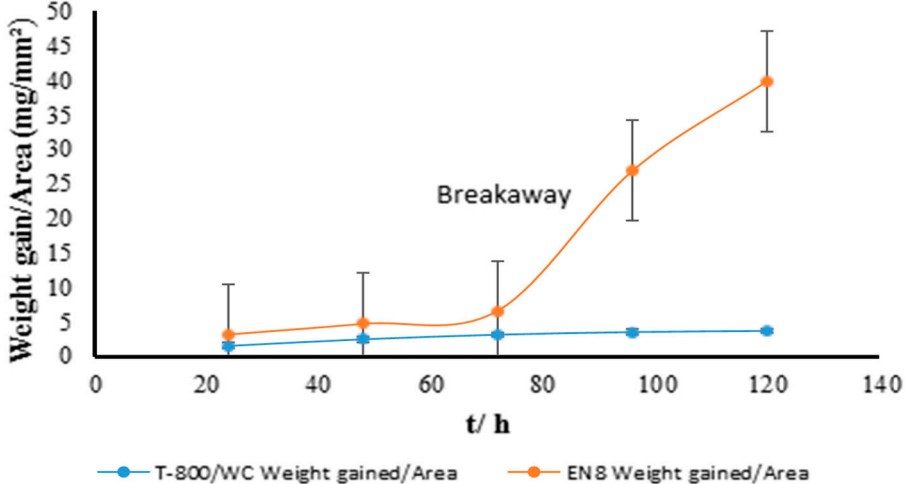

**Figure 1.** Kinetic information of the oxidation of the T-800/WC cladding coating and the substrate at 800 °C.

### 3.2. Phase Analysis of the T-800/WC Coating and EN8 Oxides

3.2.1. Phase Analysis (XRD) of EN8

Figure 2 shows XRD patterns of the oxidised EN8 substrate at 800 °C over 120 h. The main crystalline phase in the unoxidised EN8 substrate is metal iron (α-Fe: BCC crystal), whilst the final oxide phase is the hematite (Fe$_2$O$_3$) In similarity to findings from Lysenko et al. [22], analysis of Figure 2 suggests that the oxidation of the EN8 substrate proceeds according to Equations (3)–(5). In Equations (3) and (4), the oxidation of α-Fe results in the simultaneous production of iron (II) oxide and magnetite (Fe$_3$O$_4$). Thereafter, the unstable magnetite is oxidised further, and/or the iron is oxidised to hematite (Fe$_2$O$_3$), as represented in Equation (5).

$$2Fe + O_2 \; \rightarrow \; 2FeO \tag{3}$$

$$6FeO + O_2 \; \rightarrow \; 2Fe_3O_4 \tag{4}$$

$$4Fe_3O_4 + 2O_2 \; \rightarrow \; 6Fe_2O_3 \tag{5}$$

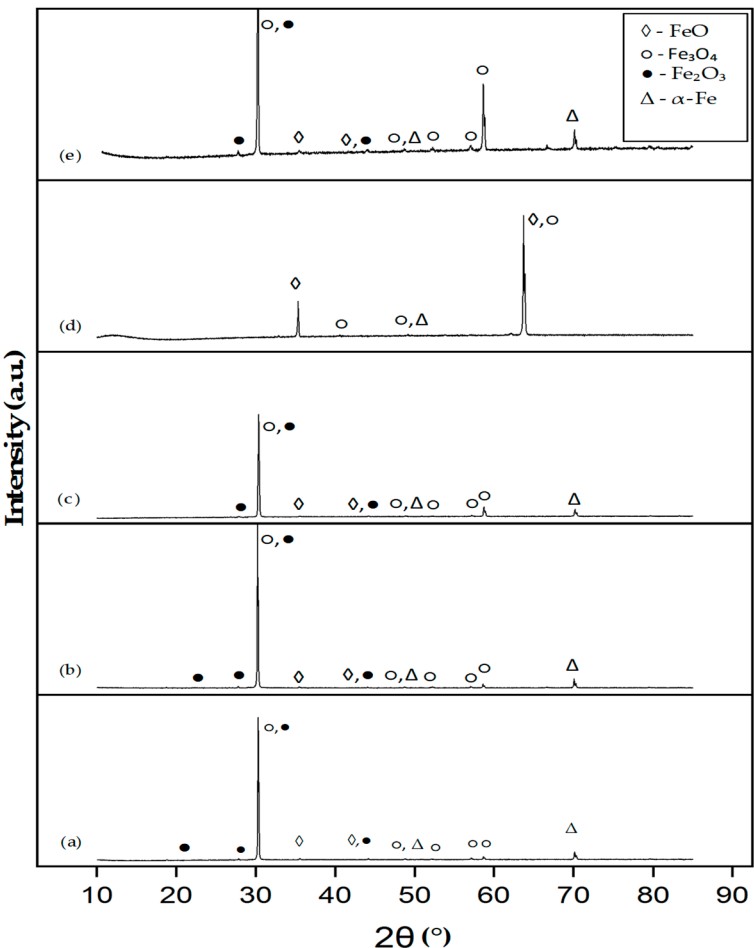

**Figure 2.** XRD patterns for EN8 after (**a**) 24, (**b**) 48, (**c**) 72, (**d**) 96 and (**e**) 120 h.

According to Figure 2a, there is a higher intensity of iron (II) oxide (FeO) at t = 24 h compared to when t = 48 and 72 h due to the reaction of iron ($\alpha$-Fe) with oxygen. When t = 48 h (Figure 2b), there are higher intensities of magnetite ($Fe_3O_4$) emanating from the further reaction of unstable iron (II) oxide (FeO) with oxygen. At t = 96 h (Figure 2d), however, there is a resurgence of higher intensities of iron (II) oxide (FeO), which could be attributed to the exposure of the EN8 base material due to the cracking phenomenon. At t = 96 h (Figure 2d), there is an increased presence of $Fe_3O_4$ as the exposure time of EN8 material increases. With increased exposure at t = 120 h, all three oxide types are present. These oxides will continue to grow until the internal stresses reach the maximum before cracking resumes. In summary, it should be noted that the complex oxidation reaction of the uncoated EN8 substrate cannot be simply elucidated with a single step, as depicted in Equations (3)–(5). Consequently, further studies should explore a multistep model-fitting approach for elucidating the mechanism of formation of the oxide scales on the EN8 substrate.

### 3.2.2. Phase Analysis (XRD) of the T-800/WC Coating

The XRD pattern for the T-800/WC composite coating before and after oxidation is shown in Figure 3. Figure 3 reveals that the T-800/WC composite coating produced by LCAP consists mainly of the hard laves phase, the Co-rich eutectic phases and the WC, which are embedded in the matrix.

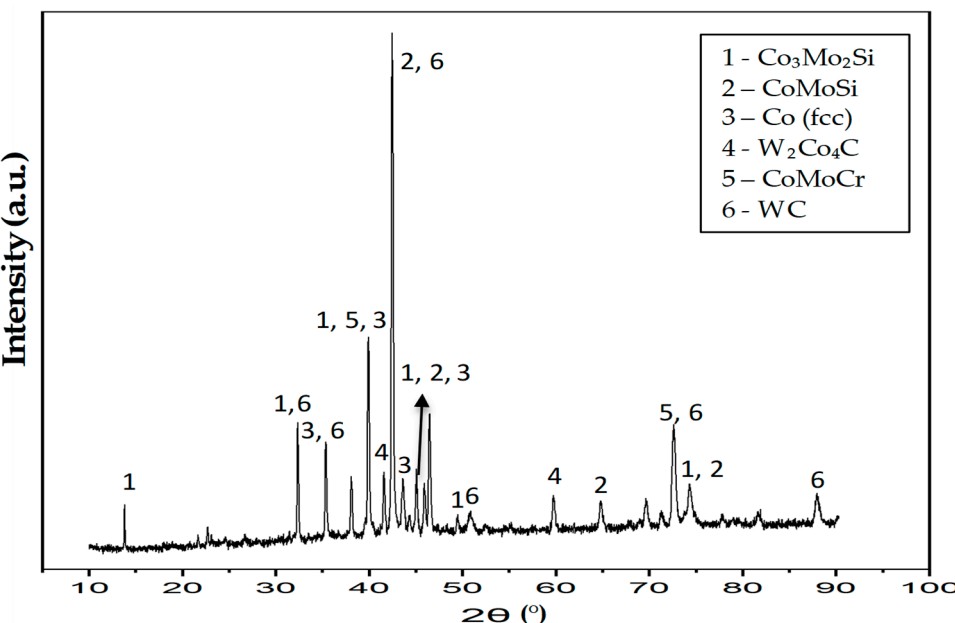

**Figure 3.** XRD pattern for the T-800/WC coating before oxidation.

The XRD patterns for the T-800/WC composite coating after oxidation (Figure 4) reveal that the oxidation process yielded laves, $Cr_2O_3$, spinel $CoCr_2O_4$ (spinel), complex oxide $CoMoO_4$, $SiO_2$ and $WO_x$ [4,21,23]. The presence of the laves phase is due to the thin oxide layer formed on the surface of the coating. The visibility of the laves phase is seen to decrease with an increase in oxidation time. This could be attributed to an increase in the oxide film covering the coating, reducing the identification of the laves phase. At oxidation time t = 24 h, there is a higher presence of complex oxide $CoMoO_4$. However, due to its instability, increased exposure makes it react and form more stable oxides, hence the reduced presence at t = 120 h. At 120 h, more stable $Cr_2O_3$ is observed.

### 3.3. Surface Morphologies and Chemical Composition of Oxide Scales on the Uncoated and EN8 Substrates

3.3.1. Surface Morphology of the Uncoated EN8 Substrate

At t = 24 h, physical observation on the oxidised uncoated EN8 revealed a velvet maroon iron oxide layer. Figure 5 shows the surface morphology of the uncoated EN8 substrate at 800 °C as the oxidation time increases. It can be seen that at t = 48 h (Figure 5a), there is a porous coating, which is because of oxygen ions permeating through the oxide layer similar to the outcome of Benitez [24]. As seen in Figure 5b, the oxide layer is seen to bulge, probably due to increasing stress because of the increase in the oxide layer thickness [22]. The external cracking phenomenon on the oxide is observed at t = 72 h which propagated due to spallation of the oxide film, as observed at t ≥ 72 h, where there is more cracking of the oxide layer exposing the substrate material to be oxidised, leading to an upsurge of mass gain per area, as shown in Figure 1. At t = 120 h, porosity is observed in the EN8 oxide coating accompanied by cracks (Figure 5e). After cracking and spallation at t = 72 h, the EN8 base material is exposed, making it susceptible to oxidation. The extent of oxide scale adhesion is affected by oxide layer thickness, the stress in the oxide layer and the rate of growth, amongst others.

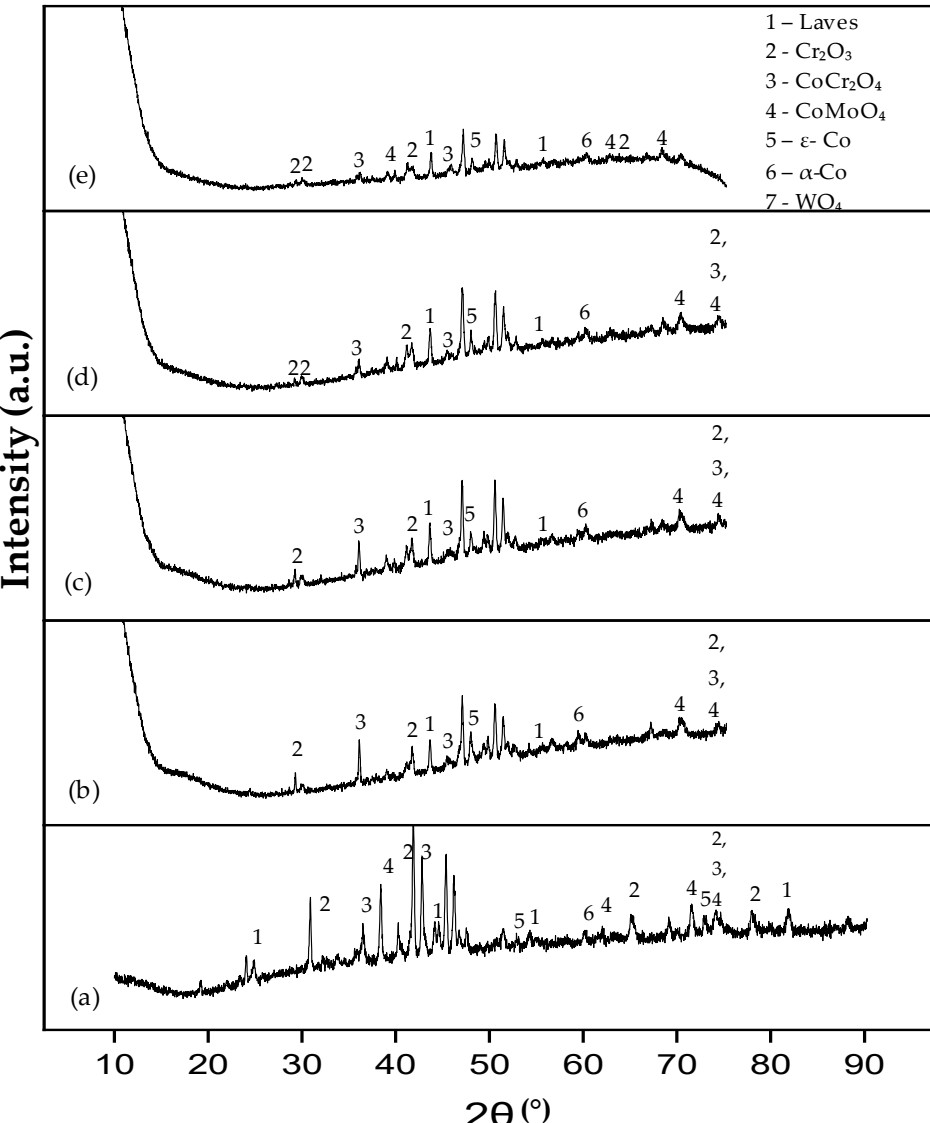

**Figure 4.** XRD patterns for the T-800/WC oxidised specimen at (**a**) 24, (**b**) 48, (**c**) 72, (**d**) 96 and (**e**) 120 h.

The cracking phenomenon exhibited in Figure 5c was further explored. Figure 6a shows that the oxide thickness grew to an average of 643.85 µm after 24 h. Internal cracking and pores can be observed in the oxide layer due to the oxide layer growth. This led to the external cracking observed in Figure 5c. Figure 6b shows the oxide thickness with an average thickness of 127.75 µm after 120 h. A new oxide layer was in the process of developing after the cracking of the initial layer at t > 72 h.

Magnified Surface View

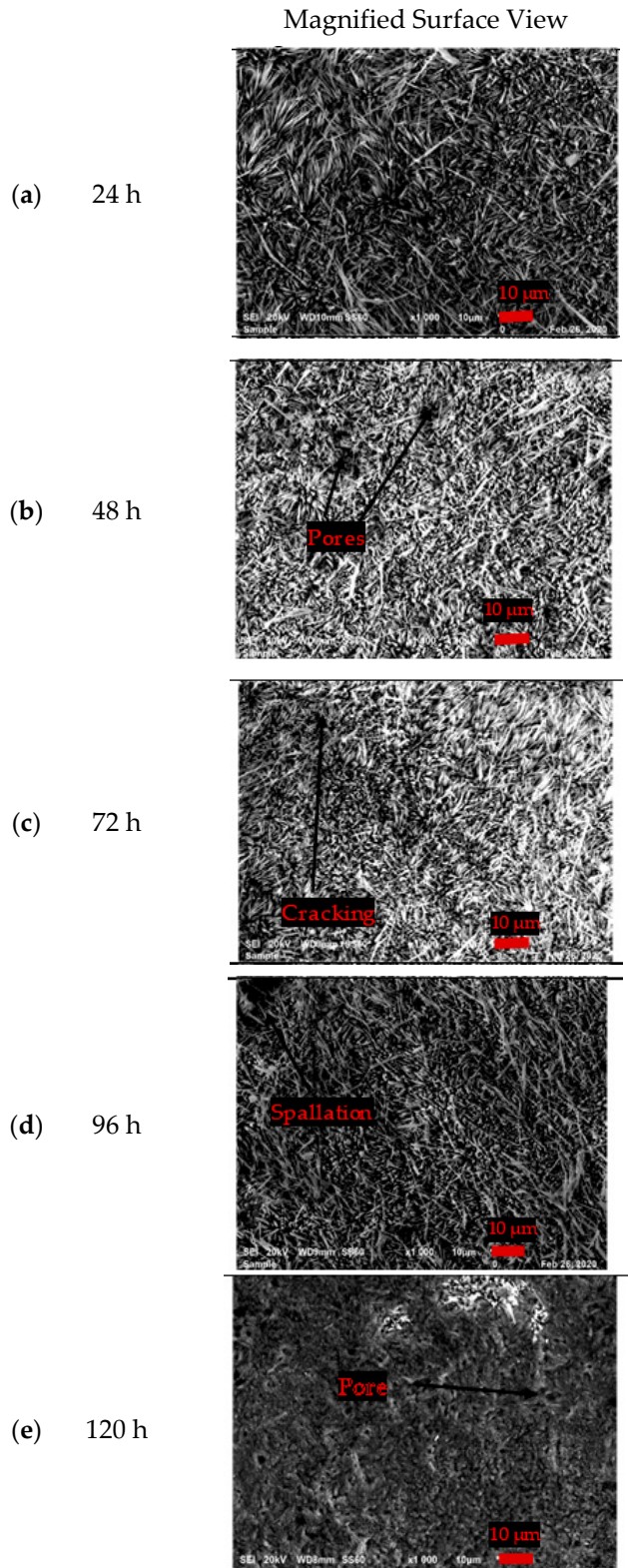

**Figure 5.** Surface morphology of EN8 for (**a**) 24, (**b**) 48, (**c**) 72, (**d**) 96 and (**e**) 120 h.

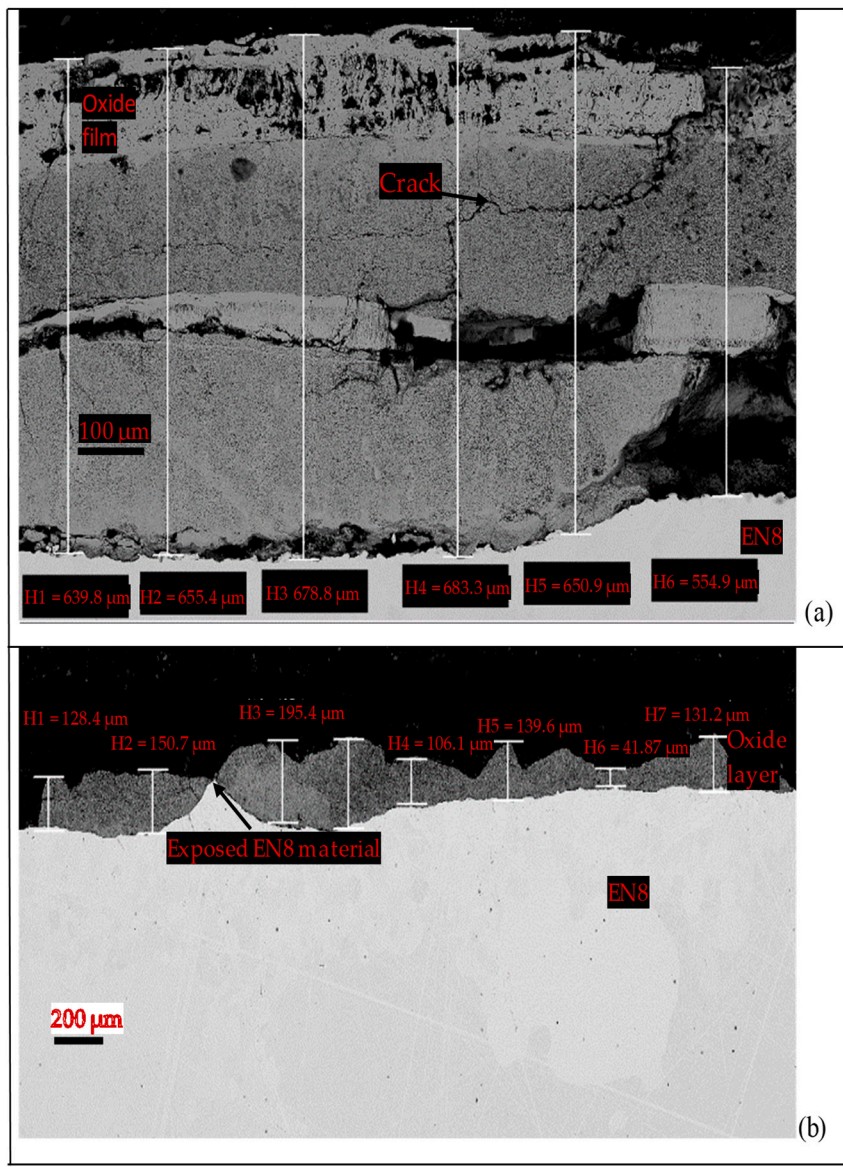

**Figure 6.** EN8 oxide film thickness at (**a**) t > 24 h and (**b**) t = 120 h.

### 3.3.2. Comparative Analysis of the Cross-Sectional View and EDS Analysis of EN8 and the T-800/WC Coating

A cross-sectional view of an oxidised EN8 material at t = 24 h is shown in Figure 7. The main constituents of the oxide layer are Fe, O and C, as revealed by the EDS analyses in Figure 4a. The oxide thickness is divided into four sections, namely (a), (b), (c) and (d). Sections (a) and (b) mainly comprise metal-rich FeO and $Fe_3O_4$ (constituents are shown in Figure 7b), which formed as an initial reaction of EN8 and oxygen. Increased exposure resulted in the further reaction of FeO to $Fe_3O_4$. Section (b) mainly comprises $Fe_3O_4$, Section (c) has a mixture of FeO and $Fe_3O_4$ and Section (d) is made up of the final stable $Fe_2O_3$. Section (d) ($Fe_2O_3$) is due to the reaction between $Fe_3O_4$ and $O_2$. This is in confirmation of the EDS finding and what other researchers also discovered that at temperatures >570 °C, the iron oxide scale is made up of FeO (wüstite), $Fe_3O_4$ (magnetite) and $Fe_2O_3$ hematite), respectively, in that order of formation on the substrate [25,26].

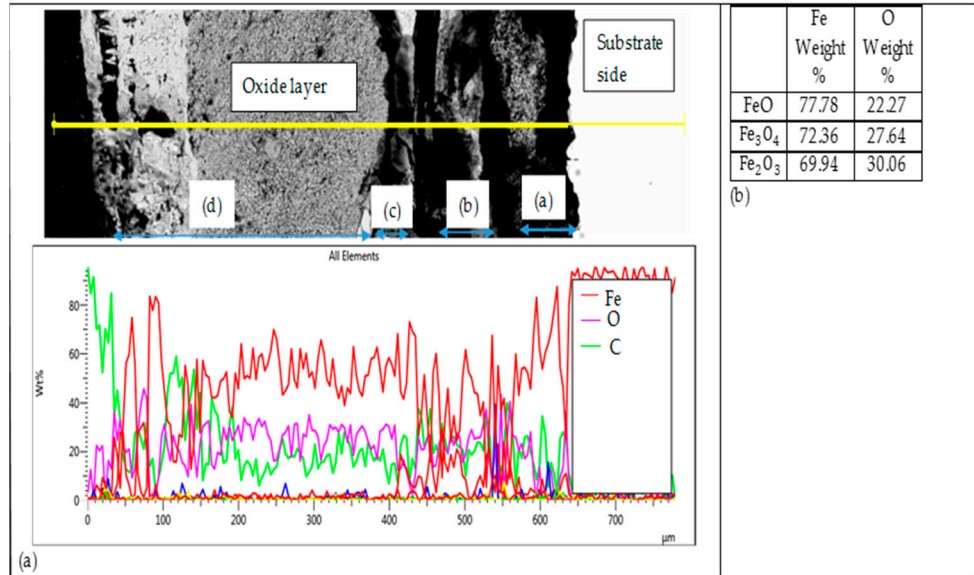

**Figure 7.** (**a**) Cross-sectional view EN8 oxide thickness and (**b**) wt.% constituents of the major composition of the oxides.

Further, an EDS line scan was carried out as shown in Figure 8 in the region marked 'oxide region' to gain more understanding of the T-800/WC coating oxide (Figure A2 shows an exploded view of the EDS).

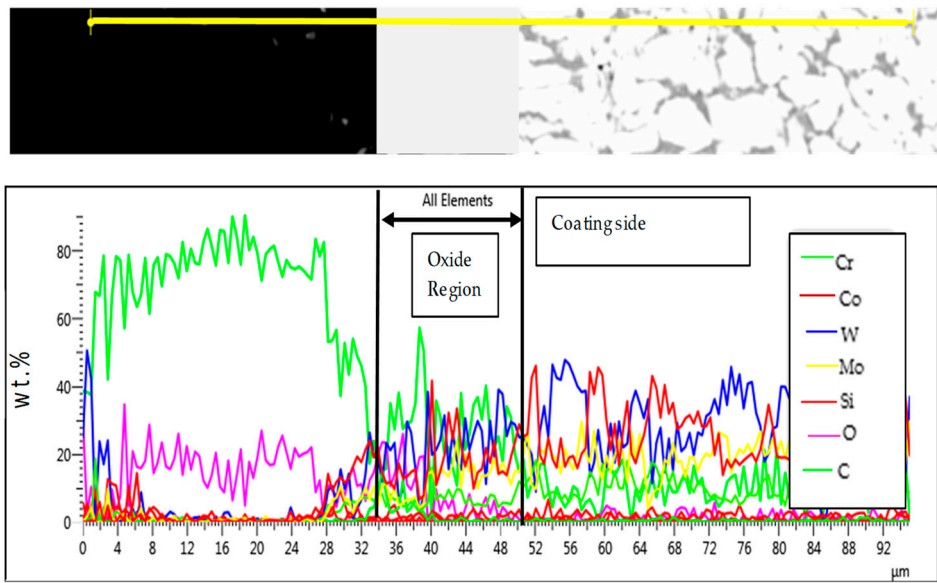

**Figure 8.** Cross-sectional view and EDS line scan of the T-800/WC specimen after oxidation at t = 120 h.

The oxide interior exhibits a high presence of Co, W, Mo, Cr, Si and oxygen. On the other hand, the oxide exterior is seen to have similar major elements (Co, W, Mo, Cr, Si and O). It can be observed that the elemental composition is higher in the interior oxide layer compared to the exterior oxide layer. With an amalgamation of this information with Figure 4, it can be deduced that the oxide layer consists of $CoMoO_4$ (with fewer contents of $WO_x$, $Cr_2O_3$ and $SiO_2$), whilst the inner layer consists of $CoMoO_4$, $WO_4$, $Cr_2O_3$ and $SiO_2$. The Co-rich solid solution $Co_{ss}$ was detected because of the very thin oxide layer that formed on the coating revealing the T-800/WC coating. It is the combination of $Cr_2O_3$ and $SiO_2$ in the interior that provides a barrier that prevents further oxidation on the

coating [23]. This is similar to findings by Peng et al. [4], who investigated the isothermal behaviour of Tribaloy T-800.

### 3.3.3. T-800/WC Surface Morphology and EDS

Physical observation of the oxidised T-800/WC coating at t = 24 h revealed a very thin greyish (almost transparent) film. SEM images of surfaces and EDS analyses of the T-800/WC coating at 800 °C are presented in Figure A1. The coating was seen to maintain a high integrity without any trace of spallation throughout the oxidation process cycles, similar to findings by Yuduo et al. [23], who investigated the effect of adding rhenium to the isothermal performance of T-800. This can be attributed to the formation of a protective stable $Cr_2O_3$, which prevented the further oxidation of the coating. From the EDS analysis, it can be concluded that O and Cr are seen to increase with an increase in oxidation time. At 120 h, the only major elements present are O, Cr and Si. This alludes to the formation of $Cr_2O_3$ and $SiO_2$ as the final oxide layer that forms on the T-800/WC LCAP coating. Initially, Co, W are seen to form oxides; however, as the oxides are not stable, further exposure to oxidation leads to further reaction.

The oxide film thickness of T-800/WC is shown in Figure 9. As seen in Figure 9a, a very thin oxide film (which could not be measured) formed on the T-800/WC coating.

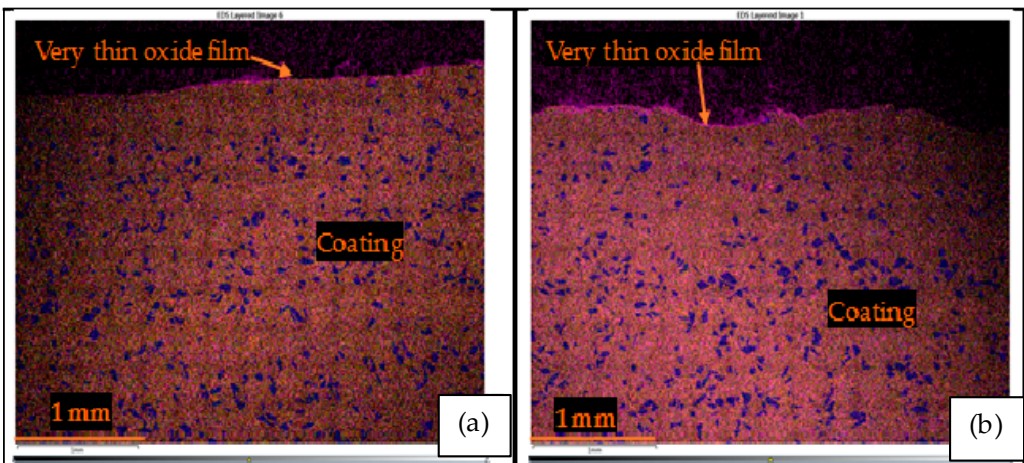

**Figure 9.** T-800/WC oxide film thickness at (**a**) t > 24 h and (**b**) t = 120 h.

However, at t = 120 h, the average oxide thickness is 13.54 μm (approximately 48 times less the oxide thickness of EN8 at t = 24). This enabled the coating to maintain its integrity even after 120 h and further confirms the superiority of the coating to high-temperature oxidation resistance as compared to EN8.

### 3.4. Oxidation Kinetics of the T-800/WC Coating and EN8

Oxidation kinetics analysis for T-800/WC and EN8 was implemented to investigate and distinguish further their oxidation behaviour. The high-temperature oxidation for the alloy mass gain is shown in Equation (6).

$$\Delta\left(\frac{M}{A}\right)^n = K_P T \tag{6}$$

where $\Delta(M/A)$ is the mass gain per unit area, $K_P$ the parabolic rate constant, T is the oxidation time and $n$ is the exponential rate. To determine the parabolic rate constant $K_P$ and the exponential rate $n$, a linear relationship plot was made for natural logarithms, ln $(M/A)$ and ln T (Figure 10). A linear relationship plot was adopted because of the low mass gain per unit area of the T-800/WC coating during oxidation [27]. The line gradient was the $K_P$ constant, whilst the line intercept was used to calculate $n$. For $n = 1$, it means it is a

kinetic linear model, whilst $n = 2$ represents a parabolic rate (which is expected for alloys). These values are recorded in Table 2.

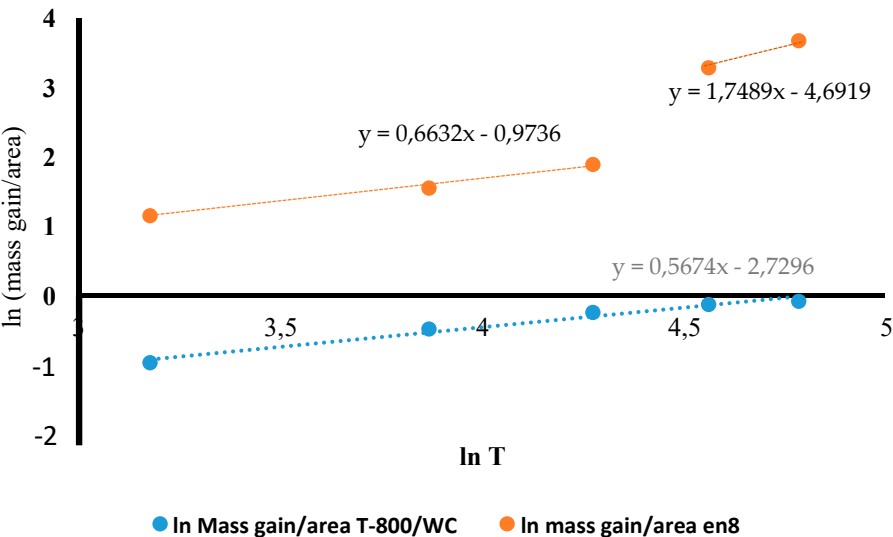

**Figure 10.** Plot of ln (mass gain/area) vs. ln T for the oxidation of EN8 and the T-800/WC coating at 800 °C for up to 120 h.

**Table 2.** $Kp$ constants for EN8 and T-800/WC.

|  | $n$ **Value** | $K_p$ **Constant/( g$^2$·cm$^{-4}$·s$^{-1}$)** |
|---|---|---|
| EN8 substrate material | 0.63 | $6.72 \times 10^{-12}$ |
| T-800/WC coating | 1.8 | $8.1 \times 10^{-13}$ |

The derived equation for T-800/WC oxidation is given in Equation (7).

$$y = 0.5674x - 2.7296 \tag{7}$$

The derived equation for the substrate (EN8) is also given in Equation (8) (before breakaway) and (9) (after breakaway).

$$y = 0.663x - 0.9736 \tag{8}$$

$$y = 1.7489x - 4.6919 \tag{9}$$

The $K_P$ constant (Table 2) shows a measure of weight gain resistance.

The $K_P$ constant of the T-800/WC coating is up to ten times better than EN8. This shows an increase in the resistance of the coating compared to the substrate. The exponential constant for EN8 of 0.63 can be rounded off to 1, representing a linear kinetic model, whilst that of T-800/WC of 1.8 can be rounded off to 2 symbolic of a parabolic rate. These results can be also linked to the surface morphology representation in Figures 5 and A1 of EN8 and the T-800/WC coating, respectively. A paralinear oxidation law for EN8 is characterised by the formation of a thick oxide scale accompanied by spallation (Figure 5). On the other hand, a parabolic law for the T-800/WC coating is characterised by moderately thin impermeable oxide with no spallation (Figure A1). This is further proof that T-800/WC has relatively higher high-temperature oxidation resistance than EN8.

*3.5. Mechanism of Elevated-Temperature Oxidation Resistance of the EN8 Substrate When Modified with the T-800/WC Coating*

Surface EDS analysis of Spot A for T-800/WC at t = 24 h is shown in Figure 11. It can be seen that Cr, O$_2$ and Si on the coating surface increase with increased exposure time.

Furthermore, the percentage (%) composition of Co, W and other minor elements is seen to decrease with time. This infers that the protective layer formed on the coating surface inhibits the further reaction of other elements (Co, W, and Mo) with oxygen, hence the depletion depicted in Figure 11.

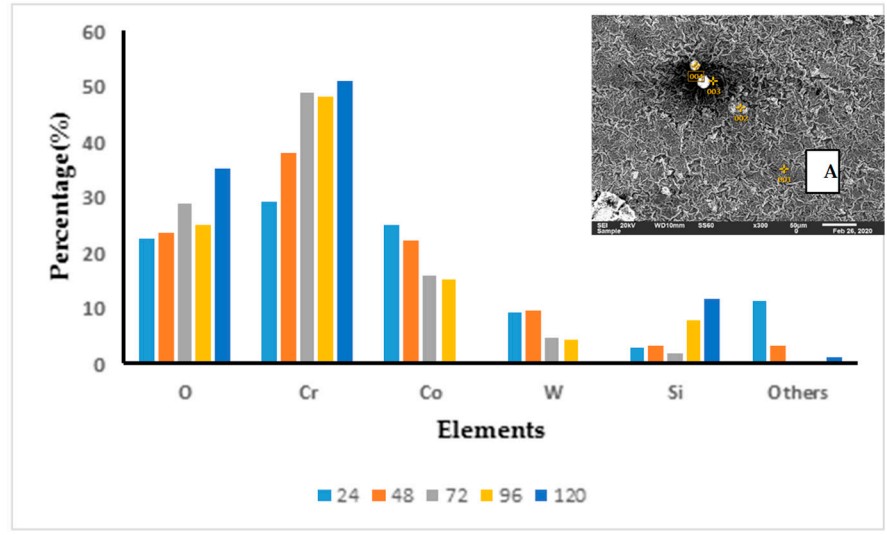

**Figure 11.** Elemental EDS analysis of the surfaces of the T-800/WC coating for up to 120 h.

The trend observed in Figure 11 can be explained by the oxidation mechanism developed for T-800/WC (Figure 12). It was observed that there is diffusion of coating ions from its surface to the oxide film/atmosphere interface where it reacts with oxygen. As the oxide layer thickness increases, the diffusion of the coating ions reduces the oxidation (oxide growth) process. The increasing oxygen content up to t = 120 h is because of reduced coating elements available for reaction (Figure 11). A reduced content of Co, W and other elements at t ≥ 48 h can be attributed to their reduced diffusion as $Cr_2O_3$ oxide thickness increases with time.

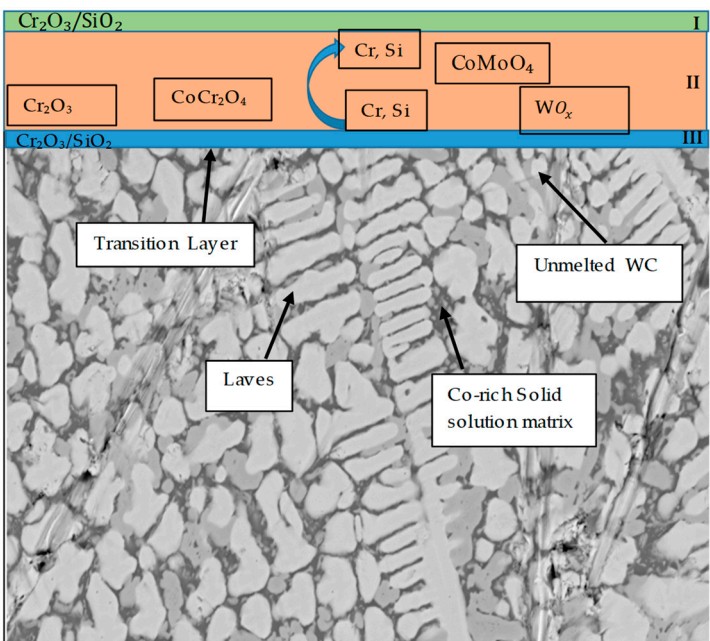

**Figure 12.** The oxidation mechanism of the LCAP T-800/WC coating.

Figure 12 shows the oxidation mechanism of the T-800/WC coating. This was drawn from a combination of XRD (Figure 3) pattern and EDS analysis.

Three regions were identified. The outer layer (I) mainly comprises $Cr_2O_3$ and $SiO_2$. The intermediate layer (II) is a mixture of $Cr_2O_3$, $SiO_2$, $CoMoO_4$, $CoCr_2O_4$ and $WO_4$. Finally, a transition discontinuous layer (III), consisting mainly of $Cr_2O_3$ and $SiO_2$ embedded in the laves, is observed. Due to low $\Delta G$, oxidation according to Equations (10)–(12) readily takes place. This then forms a layer over the coating. The $Cr_2O_3$ layer growth is due to the diffusion of chromium to the surface (Figure 12).

$$Si + O_2 = SiO_2 \quad \Delta G = -710 \text{ kJ/mol} \tag{10}$$

$$\tfrac{4}{3}Cr + O_2 = \tfrac{2}{3}Cr_2O_3 \quad \Delta G = -565 \text{ kJ/mol} \tag{11}$$

$$2Co + O_2 = 2CoO \quad \Delta G = -320 \text{ kJ/mol} \tag{12}$$

It can thus be deduced that a layer of $SiO_2/Cr_2O_3$ will initially form on the surface of the T-800/WC coating. The other oxides then form, as highlighted Figure 12 ([15,23] dos Nascimento et al. and Yuduo et al.). Finally, a layer of $SiO_2/Cr_2O_3$ formed.

## 4. Conclusions

This work described the oxidation behaviour of EN8 and T-800/WC at 800 °C. The results reveal that EN8 stainless steel (substrate) exhibited higher-temperature oxidation compared to the T-800/WC coating. The findings of this work are as follows:

The mass gain per unit area of the T-800 composite coating was eight times less than that of the uncoated EN8 substrate.

- The $Kp$ constants for the coating were found to be approximately ten (10) times lower than for EN8, showing an increase in high-temperature resistance. The parabolic constant $(K_p)$ for EN8 was $6.72 \times 10^{-12}$ $g^2 \cdot cm^{-4} \cdot s^{-1}$, whilst that for the T-800 composite coating was $8.1 \times 10^{-13}$ $g^2 \cdot cm^{-4} \cdot s^{-1}$. This was attributed to a stable chromium oxide $(Cr_2O_3/SiO_2)$ layer that formed on the surface of the coating, thereby preventing further oxidation, whilst the iron oxide film that formed on the EN8 substrate allowed the permeation of the oxygen ions into the oxide.
- The iron oxide $(Fe_2O_3)$ film that developed on EN8 spalled, as evidenced by the cracking of oxide when the oxidation time was greater than 72 h, whilst the $Cr_2O_3$ film maintained its integrity at 120 h.
- A parabolic law was observed by the T-800 composite coating, whilst a paralinear law was reported for EN8 (as evidenced by *n*-values of 0.63 and 1.8, respectively) at 800 °C up to 120 h. It can thus be deduced that this coating can be used in turbine parts where temperatures are <800 °C.

**Author Contributions:** Conceptualization, S.T.N.; methodology, S.T.N., E.O.O. and S.L.P.; software, S.T.N.; validation, S.T.N., S.L.P. and E.O.O.; formal analysis, S.T.N.; investigation, S.T.N.; resources, S.L.P. and E.O.O.; data curation, S.T.N.; writing—original draft preparation, S.T.N.; writing—review and editing, E.O.O. and S.L.P.; visualization, S.T.N.; supervision, E.O.O. and S.L.P.; project administration, E.O.O.; funding acquisition, E.O.O. and S.L.P. All authors have read and agreed to the published version of the manuscript.

**Funding:** This research was funded by the African Laser Centre (ALC) under Grant No. CSIR-NLC Reference LHIL 500 task ALC R014 together with Research Initiation Fund (Grant No. BIUST/ds/r&I/7/2016) provided by Botswana International University of Science and Technology.

**Institutional Review Board Statement:** Not applicable.

**Informed Consent Statement:** Not applicable.

**Data Availability Statement:** Data is contained within the article.

**Conflicts of Interest:** The authors declare no conflict of interest.

## Appendix A

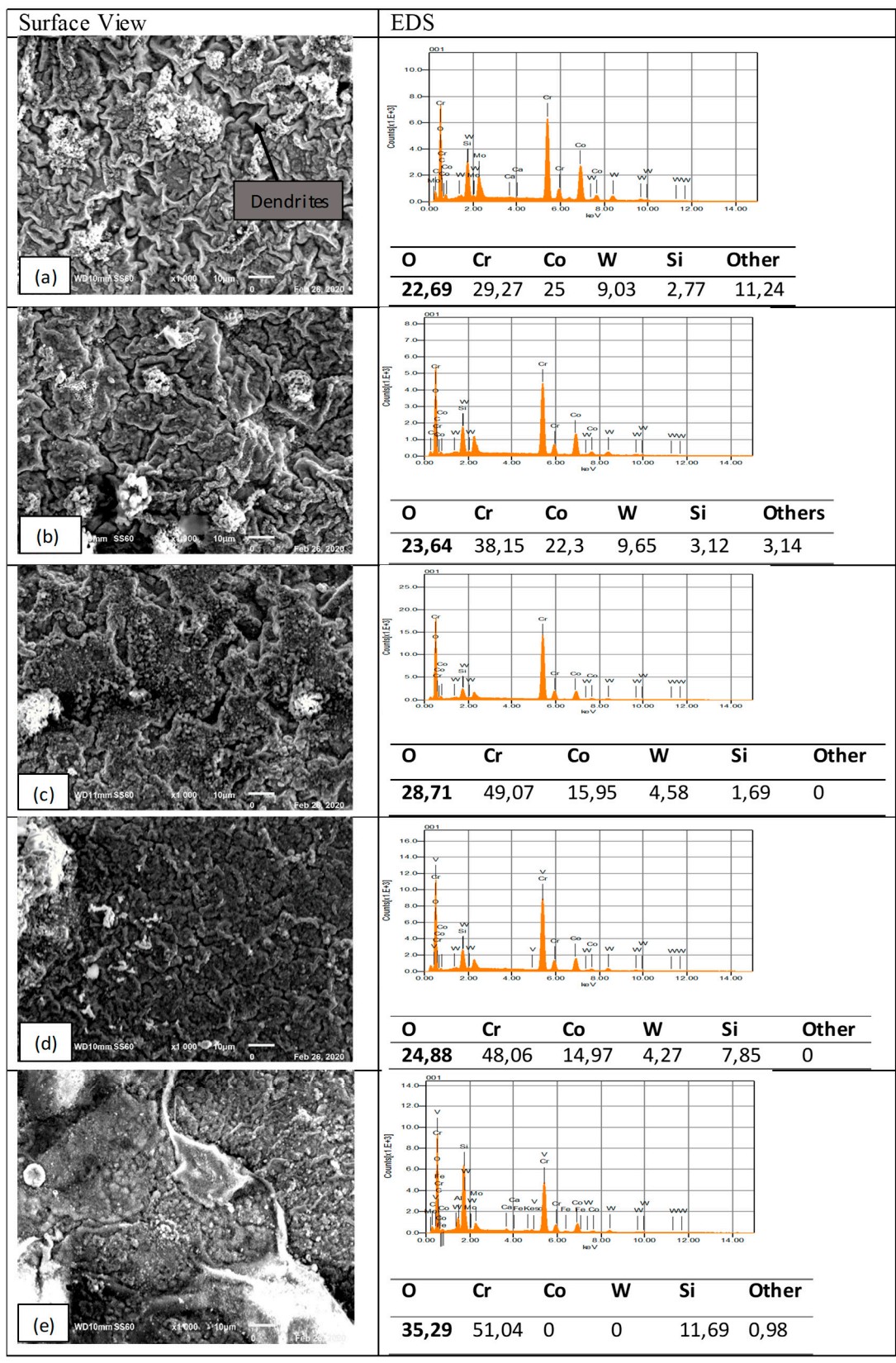

**Figure A1.** T-800/WC surface view and EDS at (**a**) 24 h, (**b**) 48 h, (**c**) 72 h, (**d**) 96 h and (**e**) 120 h.

## Appendix B

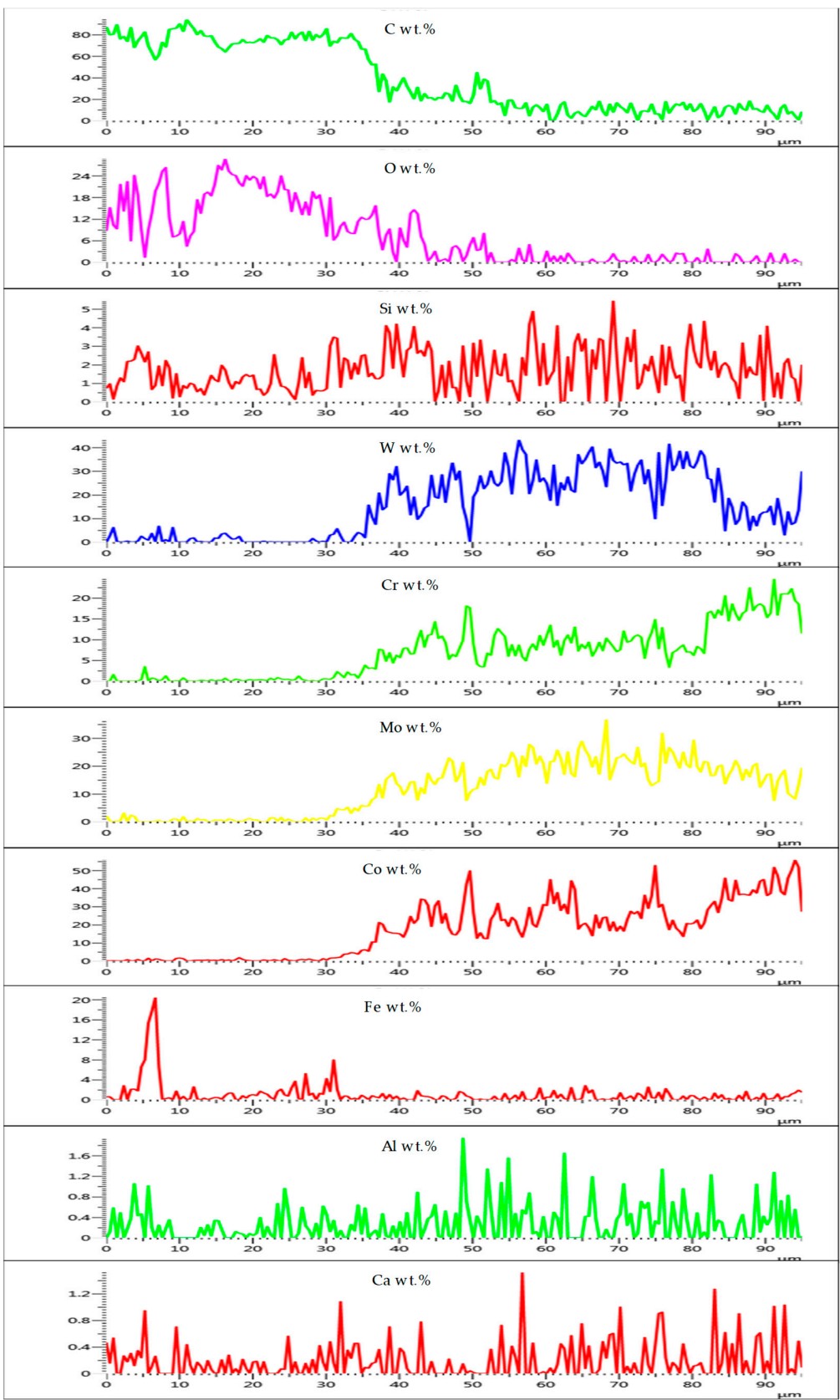

**Figure A2.** Exploded EDS analysis of T-800/WC oxide.

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
