# Peer review of "Isothermal Oxidation Performance of Laser Cladding Assisted with Preheat (LCAP) Tribaloy T-800 Composite Coatings Deposited on EN8"

_coatings, doi:10.3390/coatings11070843_

Round 1
Reviewer 1 Report
This paper seems to contain the meaningful experimental results and mathematical evaluations about the isothermal oxidation of LCAP composite coatings on stainless steel.
However, there are some figures that are difficult to recognize in the experimental results.
It is better to reconstruct some figures and related text.
Specific comments.
As for Figure2
L175: Figure 11d → Figure 2d?
The numbers in the figure are crushed and difficult to recognize.
Please indicate the diffraction peak position of α-Fe. Since the weight increase is small up to 72 hours, can you think that the strong diffraction peak is α-Fe?
Figure3
The numbers in the figure are crushed and difficult to recognize.
The indexes of (a) and (b) to (f) are different. Will the WC coating be undetectable after heating?
Fugure4
The scale of the photo can't be read.
Please explain what kind of phenomenon is occurring in (d) where there is no quantification of C and in (e) there is a rapid increase. Are carbides detected as a result of X-ray diffraction?
Figure5
The scale of the photo can't be read.
Are (a) 24h and (b) 120h correct?
As for Figure6
L241: Is the evaluation of the oxide film for 24 hours correct?
Figure7
The scale of the photo can't be read.
Figure8
What do the yellow and blue lines in the photo indicate?
The index in the figure below overlaps with the line and cannot be read.
Author Response
Reviewer #1
(x) I would not like to sign my review report
( ) I would like to sign my review report
English language and style
( ) Extensive editing of English language and style required
( ) Moderate English changes required
(x) English language and style are fine/minor spell check required
( ) I don't feel qualified to judge about the English language and style
|
Yes |
Can be improved |
Must be improved |
Not applicable |
|
|
Does the introduction provide sufficient background and include all relevant references? |
(x) |
( ) |
( ) |
( ) |
|
Is the research design appropriate? |
(x) |
( ) |
( ) |
( ) |
|
Are the methods adequately described? |
(x) |
( ) |
( ) |
( ) |
|
Are the results clearly presented? |
( ) |
( ) |
(x) |
( ) |
|
Are the conclusions supported by the results? |
( ) |
(x) |
( ) |
( ) |
Comments and Suggestions for Authors
This paper seems to contain the meaningful experimental results and mathematical evaluations about the isothermal oxidation of LCAP composite coatings on stainless steel.
- However, there are some figures that are difficult to recognize in the experimental results. It is better to reconstruct some figures and related text. Response: The Figures have been reconstructed throughout the manuscript.
- Specific comments.
As for Figure 2
L175: Figure 11d → Figure 2d? Response: Yes corrected on page 5, paragraph 1, line 5
- The numbers in the figure are crushed and difficult to recognize. Response: Corrected in Figure 2 on page 5.
- Please indicate the diffraction peak position of α-Fe. Since the weight increase is small up to 72 hours, can you think that the strong diffraction peak is α-Fe? Response: No its not, it is iron oxides which vary with increase in exposure time. The figure in the 1st submission was not clear but this has been corrected in Figure 2 on page 5 which clearly shows the peaks.
- Figure 3
The numbers in the figure are crushed and difficult to recognize.
The indexes of (a) and (b) to (f) are different. Will the WC coating be undetectable after heating? – Response: This has been rectified by splitting Figure 3 to Figures 3 and 4 on pages 6 and 7 respectively. The Figures are now clearer. Yes a WOx is detected very lightly.
- Figure 4
The scale of the photo can't be read. Response: This has been rectified as shown in Figure 5 on page 8.
- Please explain what kind of phenomenon is occurring in (d) where there is no quantification of C and in (e) there is a rapid increase. Are carbides detected as a result of X-ray diffraction? Response: Yes it could be because of X-ray diffraction. However, the EDS has since been removed based on recommendations by some one of the reviewers.
- Figure 5
The scale of the photo can't be read. Response: This has been rectified as shown in Figure 6, on page 9.
- Are (a) 24h and (b) 120h correct? Response: Yes, at a) there is oxide layer build up. At T>72 hours, there is spallation which results in a breakdown of old layer and build-up of new oxide layer, hence the thinner layer at 120 hours.
- As for Figure 6
L241: Is the evaluation of the oxide film for 24 hours correct? Response: Yes, diagram was edited according to recommendation by another reviewer as shown in Figure 7 on page 10.
- Figure 7
The scale of the photo can't be read. Response: This has been rectified by replacing the diagram with another diagram in Figure 9 on page 11 to improve visibility.
- Figure 8
What do the yellow and blue lines in the photo indicate? Response: Lines removed. A black line remains as seen in Figure 8 on page 10
- The index in the figure below overlaps with the line and cannot be read Response: Corrected
Reviewer 2 Report
see attached comments

Author Response
Reviewer #2 report
(x) I would not like to sign my review report
( ) I would like to sign my review report
English language and style
( ) Extensive editing of English language and style required
( ) Moderate English changes required
( ) English language and style are fine/minor spell check required
(x) I don't feel qualified to judge about the English language and style
|
Yes |
Can be improved |
Must be improved |
Not applicable |
|
|
Does the introduction provide sufficient background and include all relevant references? |
( ) |
( ) |
( ) |
(x) |
|
Is the research design appropriate? |
(x) |
( ) |
( ) |
( ) |
|
Are the methods adequately described? |
(x) |
( ) |
( ) |
( ) |
|
Are the results clearly presented? |
( ) |
( ) |
(x) |
( ) |
|
Are the conclusions supported by the results? |
( ) |
( ) |
(x) |
( ) |
Comments and Suggestions for Authors
see attached comments
Reviewer #2 report
General comment
The paper reports about investigations on the oxidation behaviour of carbon steel protected by Co based coating containing Cr and Si. The objective is interesting for the community. However, parts of the introduction are confusing, the quality of some figures is very low and some results of the investigations give less information about the material system but seem indicating that the authors have not understood their meaning.
Therefore, the reviewer cannot recommend accepting the paper.
Detailed comments
- The whole first part of the introduction is confusing.
Line 31: Oxidation is not a form of corrosion. Oxidation means an increase of the oxidation number, for instance for the reaction 2Fe + O2 the oxidation number of iron increases from 0 to 2. Corrosion is a form of oxidation, not opposite. Response: Corrected
Line 34/35: The sentence “High temperatures have a tendency of accelerating the phenomenon of oxidation.” is banal. Response: Eliminated
- Figure 1: t = 72 h is not the breakaway point, it is the last data point in the parabolic oxidation mode. The breakaway starts sometimes between 72 and 96 h. Response: This has been rectified as shown in Figure 1 on page 4, where a breakaway section is highlighted instead.
- Eq. 4 and 5 would be better:
6FeO + O2 = 2Fe3O4 (4)
4Fe3O4 + O2 = 6Fe2O3 (5) Response: Effected
- Figures 2, 3 EDS in A1: The quality of these figures is too low. The numbers in the diffraction patterns indicating the phase and labels of the EDS peaks are too tiny. Response: Corrected as can be observed in Figure 2 on page 5. However, to improve the quality of the figures, Figure 3 was split into Figures 3 and 4 on pages 6 and 7 respectively.
- Line 178: The α-Fe peak has vanished already after 96 h, not as described at 120 h. Response: Corrected.
- Right column of Figure 4: The data are nonsense. There is no iron oxide with a higher atomic percentage of oxygen than of iron. The values for the sample oxidized 5 days indicate that the surface consist mainly of carbon oxides. Response: Right column was removed and replaced with expanded EDS to show the element composition at 24 hours and at 120 hours.
- Eq. 6, line 301: Kp is the parabolic rate coefficient, not a constant because it depends on temperature. A parabolic rate coefficient can only by determined for n=0.5. Rate coefficients determined for other n cannot be compared for different exponents.
Response: We maintain that Kp is a parabolic rate constant derived from equation 6 calculated at constant temperature (800 °C). Several other authors have used the same name for their calculations of oxidation kinetics (Wang et al., 2020; Duan et al., 2019; Peng et al., 2018; Yuduo et al., 2010). The same authors further confirm that n is the rate exponent. From equation (6) if n =1 then the kinetic model is linear and n = 2 parabolic kinetic model (common for alloys and in this case T-800/WC (n = 1.8 ≈ 2).
- Fig. 9 has a low quality. Response: Figure 10 on page 12 replaces Figure 9. This after effecting corrections from other reviewers also.
- Eq. 8 is nonsense. As the authors has described, a change in the reaction kinetics was observed because of oxide scale breakaway. It cannot be described by one equation. Two equations has to be determined for the time before and after starting the breakaway. Response: The equations for before and after scale breakaway were determined as shown in Figure 9 and equations 8.1 – 8.2.
The data for the coated material should be fitted with n=0.5 instead of 0.5674 (Eq. 7). Response: This was not possible based on our submission in vii).
- Line 351/352: The main process of growing of a Cr2O3 layer is the diffusion of chromium to the surface, not the diffusion of oxygen to the oxide/metal interface. Response: Corrected
11 Line 359: The reviewer guess that the figure number is wrong. Response: Response: Corrected
It is not understandable why the iron oxides should occur before the oxides mentioned in line 193. Response: May this be clarified, it is not clear.
References
Duan, H., Liu, Y., Lin, T., Zhang, H., & Huang, Z. (2019). Investigation on the High-Temperature Oxidation Resistance of Ni-(3~ 10) Ta and Ni-(3~ 10) Y Alloys. Metals, 9(1), 97.
Peng, J., Fang, X., Marx, V., Jasnau, U., & Palm, M. (2018). Isothermal oxidation behavior of Tribaloy TM T400 and T800. npj Materials Degradation. 2(1), 1-7. doi:10.1038/s41529-018-0060-3
Wang, D., Li, H., & Zheng, W. (2020). Oxidation behaviors of TA15 titanium alloy and TiBw reinforced TA15 matrix composites prepared by spark plasma sintering. Journal of Materials Science & Technology, 37, 46-54.
Yuduo Z., Zhigang Y., Chi Z., Hao L. (2010). Effect of Rhenium Addition on Isothermal Oxidation Behavior of Tribaloy T-800 Alloy. Chinese Journal of Aeronautics. 23, 370-376
Reviewer 3 Report
The manuscript is devoted to the study of the oxidative behavior of a promising composite protective coating and the original EN8 steel at elevated temperatures. This work is distinguished by a strong methodological formulation of the study, which made it possible to highlight the key points in the oxidation of steel and shed light on the mechanism of hot corrosion of a complex coating. The strongest drawback of the work is poor visualization of the material under discussion, and this greatly complicates the perception.
Some comments on the manuscript:
1) In the kinetic part, the data linearization applied by the authors seems to be reasonable for T-800/WC coating, since all points fall on a straight line (Fig. 9, blue line). However, the approximation of the data for the initial EN8 steel raises a question. The bottom three points seem to belong to one approximation line, and the top two to the other approximation line (Fig. 9, orange line). Therefore, the relevance of the obtained kinetic data may be insufficient.
2) What was the thickness of the T-800/WC coating under study and what was the choice of this thickness based on?
3) The data obtained by different methods are discussed mostly separately. Please add more comparisons. For example, it is reasonable to compare the XRD and EDS data of the composition of the oxidized layers (for steel, these are lines 241-251, for coverage, these are lines 284-294).
4) lines 146-147. Figure 1 does not show that the oxidation of the coating at a holding time of <72 hours is faster than that of uncoated steel. Confirm this with values.
5) It is necessary to substantially improve all the Figures. Some specific examples:
a) Add the error in determining the value to Figure 1. Remove the extra character from the caption in the figure. Improve image quality.
b) Contrary to the text, in Figure 2 there is no diffractogram of the initial (unoxidized) steel. The x-axis units should also be added. In Figure 2d, the 2 theta measurement area is smaller than the others. The numbers denoting peaks, and their interpretation should be increased.
c) Figure 3 is far from the text discussed it. It has two decoding of reflexes, contradicting each other. In the figures 3b,c,g,f,e, many intense reflexes are not assigned to any phase. Some graphs have a 2 theta measurement area smaller than others. The numbers denoting peaks, and their interpretation should be increased.
d) Scale is not visible on all morphology images. Image captions are too small. This is especially true for the thickness of the oxidized layer, they are unreadable.
e) As for the image of the EDS data, is it reasonable to give with such a high accuracy, for example, 77.78%. What was the error in determining the content of elements? Have the calibrations been made with reference materials with a similar morphology?
f) In Figures 6 and 7, the colors that represent different elements are duplicate. This makes perception difficult. In addition, the transcript is small and overlaps with the graphs.
6) English should be checked. For example, instead of "XRD spectra" it is correct to write "XRD patterns".
Author Response
Reviewer 3# report
Comments and Suggestions for Authors
The manuscript is devoted to the study of the oxidative behavior of a promising composite protective coating and the original EN8 steel at elevated temperatures. This work is distinguished by a strong methodological formulation of the study, which made it possible to highlight the key points in the oxidation of steel and shed light on the mechanism of hot corrosion of a complex coating. The strongest drawback of the work is poor visualization of the material under discussion, and this greatly complicates the perception.
Some comments on the manuscript:
1) In the kinetic part, the data linearization applied by the authors seems to be reasonable for T-800/WC coating, since all points fall on a straight line (Fig. 9, blue line). However, the approximation of the data for the initial EN8 steel raises a question. The bottom three points seem to belong to one approximation line, and the top two to the other approximation line (Fig. 9, orange line). Therefore, the relevance of the obtained kinetic data may be insufficient. Response: The equations for before and after scale breakaway were determined as shown in Figure 9 and equations 8.1 – 8.2.
2) What was the thickness of the T-800/WC coating under study and what was the choice of this thickness based on?
Response: The thickness of the coating was 2.5 mm. However, this was an initial investigation so the objective was to determine the oxidation and oxidation mechanism of T-800/WC deposited on EN8 stainless steel. Subsequent investigations will then consider the effects of coating thickness on the oxidation and mechanism of T-800/WC coating on EN8.
3) The data obtained by different methods are discussed mostly separately. Please add more comparisons. For example, it is reasonable to compare the XRD and EDS data of the composition of the oxidized layers (for steel, these are lines 241-251, for coverage, these are lines 284-294).
Response: Executed as per the recommendation in section 3.3.2.
4) lines 146-147. Figure 1 does not show that the oxidation of the coating at a holding time of <72 hours is faster than that of uncoated steel. Confirm this with values. Response: This has been confirmed on page 4, paragraph 1, lines 5, 6.
5) It is necessary to substantially improve all the Figures. Some specific examples: Response: Improved on page 5
- a) Add the error in determining the value to Figure 1. Response: Added in Figure 1 on page 5.
Remove the extra character from the caption in the figure. Improve image quality. Response: Done.
b) Contrary to the text, in Figure 2 there is no diffractogram of the initial (unoxidized) steel. The x-axis units should also be added. In Figure 2d, the 2 theta measurement area is smaller than the others. The numbers denoting peaks, and their interpretation should be increased. Response:
c) Figure 3 is far from the text discussed it. It has two decoding of reflexes, contradicting each other. In the figures 3b, c, g, f, e, many intense reflexes are not assigned to any phase. Some graphs have a 2 theta measurement area smaller than others. The numbers denoting peaks, and their interpretation should be increased. Response: Corrected as observed in Figure 4 which was split to Figures 3 and 4 on pages 6 and 7 respectively.
d) Scale is not visible on all morphology images. Image captions are too small. This is especially true for the thickness of the oxidized layer, they are unreadable. Response: Corrected in Figure 9 on page 11.
e) As for the image of the EDS data, is it reasonable to give with such a high accuracy, for example, 77.78%. What was the error in determining the content of elements? Response: This has been corrected in Figure 7 on page 10.
Have the calibrations been made with reference materials with a similar morphology? Response: Yes
f) In Figures 6 and 7, the colors that represent different elements are duplicate. This makes perception difficult. In addition, the transcript is small and overlaps with the graphs. Response: This has been corrected.
6) English should be checked. For example, instead of "XRD spectra" it is correct to write "XRD patterns". Response: This will be done.
Round 2
Reviewer 1 Report
It is difficult to judging from XRD profiles whether the result of each phase is correct or not, so please reconfirm it.
Specific comments.
Both old and new figures are shown in Figures 2 and 4.
The new figure in Figure 2 does not have symbols such as a) b).
Since the key number at each diffraction angle is different from previous figure, it is better to reconfirm that there is no mistake.
Author Response
|
Yes |
Can be improved |
Must be improved |
Not applicable |
|
|
Does the introduction provide sufficient background and include all relevant references? |
(x) |
( ) |
( ) |
( ) |
|
Is the research design appropriate? |
(x) |
( ) |
( ) |
( ) |
|
Are the methods adequately described? |
(x) |
( ) |
( ) |
( ) |
|
Are the results clearly presented? |
( ) |
( ) |
(x) |
( ) |
|
Are the conclusions supported by the results? |
( ) |
(x) |
( ) |
( ) |
Comments and Suggestions for Authors
It is difficult to judging from XRD profiles whether the result of each phase is correct or not, so please reconfirm it.
Response:
Specific comments.
1)Both old and new figures are shown in Figures 2 and 4.
Response: Corrected
2)The new figure in Figure 2 does not have symbols such as a) b).
Response: Corrected
3) Since the key number at each diffraction angle is different from previous figure, it is better to reconfirm that there is no mistake.
Response: Reconfirmed, there is no mistake.
Reviewer 2 Report
General comment
The paper is improved marginally. Even in this version, some of the results are false.
Therefore, the reviewer cannot recommend accepting the paper in this version. Strong improves in the discussions and conclusions are needed for publishing.
Detailed comments
- Line 145, Fig. 1, Fig. 10: Use t instead T as symbol of time.
- The formula given in Eq. (4) its wrong. It must be
6FeO + O2 = 2Fe3O4 instead of 6Fe + O2 = 2Fe3O4
3. Figure 2: Improve the quality of the figure
4. Line 294, Line 311, Tab. 2, Conclusions (Lines 362 – 368): KP is only a parabolic rate coefficient if n = 0.5 otherwise it is not a parabolic one! It is not a coefficient, not a constant because it depends on temperature. Also in this framework: It makes whether a mathematical nor a physical sense comparing rate coefficients K determined for different exponents n.
5. Line 316: An exponent of 2 is for hyperbolic kinetics not for parabolic. For parabolic kinetics, n must be 0.5!
6. Line 375: remove the space between “where” and “temperatures”

Author Response
|
Yes |
Can be improved |
Must be improved |
Not applicable |
|
|
Does the introduction provide sufficient background and include all relevant references? |
(x) |
( ) |
( ) |
( ) |
|
Is the research design appropriate? |
(x) |
( ) |
( ) |
( ) |
|
Are the methods adequately described? |
(x) |
( ) |
( ) |
( ) |
|
Are the results clearly presented? |
(x) |
( ) |
( ) |
( ) |
|
Are the conclusions supported by the results? |
( ) |
( ) |
(x) |
( ) |
Comments and Suggestions for Authors
General comment
The paper is improved marginally. Even in this version, some of the results are false.
Therefore, the reviewer cannot recommend accepting the paper in this version. Strong improves in the discussions and conclusions are needed for publishing.
Detailed comments
- Line 145, Fig. 1, Fig. 10: Use t instead T as symbol of time. Response: Effected on page 4, line 13 , Figure 1 (page 4), Figure 10 (page 14)
- The formula given in Eq. (4) its wrong. It must be
6FeO + O2 = 2Fe3O4 instead of 6Fe + O2 = 2Fe3O4 Response: Corrected in equation (4) on page 5.
- Figure 2: Improve the quality of the figure: Response: Improved as shown in Figure 2
- Line 294, Line 311, Tab. 2, Conclusions (Lines 362 – 368): KP is only a parabolic rate coefficient if n = 0.5 otherwise it is not a parabolic one! It is not a coefficient, not a constant because it depends on temperature. Also in this framework: It makes whether a mathematical nor a physical sense comparing rate coefficients K determined for different exponents n.
- Line 316: An exponent of 2 is for hyperbolic kinetics not for parabolic. For parabolic kinetics, n must be 0.5!
Response to 4 and 5: We still maintain our position as submitted earlier concerning Kp being a parabolic rate constant and n (rate exponent) =2 representing parabolic oxidation. We supported this through relevant literature in our initial submission. Further analysis from the literature shows that both Kp and n are dependent on temperature (Kp is constant at a particular temperature. Our work did not vary temperature but rather time) and can vary for different materials. Similar to findings from the literature, our value of n = 1.8 (approximately 2) is parabolic. To support our position, we consider the work of Korpiola (2004) who reported that the ideal ionic diffusion-controlled oxidation of metals, follows a parabolic oxidation rate law described as shown below:
(1)
where A is the area on which the reaction occurs, m is mass, Kp is the reaction rate constant or parabolic rate constant (g exp2 cm exp -4 s exp-1) and t is time. Equation (1) can be rearranged as:
(2)
Equation (2) suggests that the reaction rate constant or parabolic rate constant is strongly affected by the surface area of the constituents of the coatings. It is clear from the analysis above that the 1/n = 0.5 is what the reviewer is erroneously referring to as n = 0.5 for the parabolic model of oxidation of metals and alloys.
References:
- Korpiola (2004) High temperature oxidation of metal alloy and cermet powders in HVOF spraying process. PhD thesis, Department of Materials Science and Rock Engineering, Helsinki University of Technology, Espoo, Finland. 109p.
- Line 375: remove the space between “where” and “temperatures”. Response: corrected on page 16 line 19
